# A Robust Detection Algorithm for Infrared Maritime Small and Dim Targets

**DOI:** 10.3390/s20041237

**Published:** 2020-02-24

**Authors:** Yuwei Lu, Lili Dong, Tong Zhang, Wenhai Xu

**Affiliations:** School of Information Science & Technology, Dalian Maritime University, 1 Linghai Road, Ganjingzi District, Dalian 116033, China; luyw@dlmu.edu.cn (Y.L.); zhangt@dlmu.edu.cn (T.Z.); wenhaixu@dlmu.edu.cn (W.X.)

**Keywords:** infrared maritime target detection, small dim target, median filter, gradient feature, adaptive segmentation

## Abstract

Infrared maritime target detection is the key technology of maritime target search systems. However, infrared images generally have the defects of low signal-to-noise ratio and low resolution. At the same time, the maritime environment is complicated and changeable. Under the interference of islands, waves and other disturbances, the brightness of small dim targets is easily obscured, which makes them difficult to distinguish. This is difficult for traditional target detection algorithms to deal with. In order to solve these problems, through the analysis of infrared maritime images under a variety of sea conditions including small dim targets, this paper concludes that in infrared maritime images, small targets occupy very few pixels, often do not have any edge contour information, and the gray value and contrast values are very low. The background such as island and strong sea wave occupies a large number of pixels, with obvious texture features, and often has a high gray value. By deeply analyzing the difference between the target and the background, this paper proposes a detection algorithm (SRGM) for infrared small dim targets under different maritime background. Firstly, this algorithm proposes an efficient maritime background filter for the common background in the infrared maritime image. Firstly, the median filter based on the sensitive region selection is used to extract the image background accurately, and then the background is eliminated by image difference with the original image. In addition, this article analyzes the differences in gradient features between strong interference caused by the background and targets, proposes a small dim target extraction operator with two analysis factors that fit the target features perfectly and combines the adaptive threshold segmentation to realize the accurate extraction of the small dim target. The experimental results show that compared with the current popular small dim target detection algorithms, this paper has better performance for target detection in various maritime environments.

## 1. Introduction

Infrared imaging systems has been an essential part of maritime target searching systems for some time now [1], so that detecting maritime targets using infrared images is a crucial technique [2]. However, in the infrared images, the texture and structure information of the target is lacking, and the small dim targets often occupy only a few pixels, which makes the small dim infrared targets often appear as bright spots. In addition, due to the temperature change, the contrast between the target and the background is not easy to distinguish, and the target may be lost due to the hot cross at any time of the day [3]. That is, the target is often submerged in the background submerged in complex background [4].

In general, infrared small target detection methods can be divided into two types based on sequence images and single-frame images. The sequence-based method needs to capture the target motion trajectory through continuous multiple frames of images, judge and eliminate interference [5]. When the background is smooth and the target is moving slowly, Robinson Guard spatial filter [6], dynamic programming method [6], and three-dimensional matched filtering [7] can all have good results. However, in practical applications, due to factors such as the jitter of the imaging platform, it is difficult to guarantee the homogeneity of the background, and when the target moves fast, the detection effect of these algorithms will decrease sharply. At the same time, the high storage space and hardware performance requirements make this kind of algorithm difficult to apply to actual search work. The detection based on single frame is a popular method in recent years, which has caused more and more research and attention.

Traditional single frame detection algorithms mostly focus on predicting the background information of the image, and then detect the target by the difference from the input image. The main method is to use various filtering methods for target detection. The filtering methods can be divided into spatial domain filtering and transform domain filtering. Space-based filtering includes morphological filtering [8], two-dimensional minimum mean square adaptive filter [9], and maximum median and maximum mean filter [10]. Inspired by mathematical morphological methods, Top-Hat transform is widely used in the field of image processing. [11] first used it for small target detection, and later [12] used two different but related structural elements to construct new Top-Hat transform achieves better results when performing target detection, compared to classic algorithms. This type of method has a fast processing speed, but is too sensitive to high frequency interference, and it is easy to retain too many false alarms. Transform domain-based filtering includes classic high-pass filtering in the frequency domain [13], wavelet filtering [14], and so on. This type of method mainly distinguishes the low-frequency part from the high-frequency part through various transformations, and considers the low-frequency part as the background and the high-frequency part as the target. However, in practical applications, the presence of strong high-frequency interference will greatly affect the detection performance of this type of method. Even if the reliability is improved compared to the filtering based on the spatial domain, due to the forward and inverse transformation of the image, the amount of calculation is very large, which makes it difficult for such methods to meet real-time requirements.

In recent years, methods based on the human visual system have achieved great success in detecting small infrared targets. The key of this method is the acquisition of salient regions. The most important of these methods is the method based on local contrast. In [4], the earliest method based on local contrast measurement (LCM) was proposed to detect the target by calculating the brightness difference between the target and the adjacent area. On this basis, many articles have proposed improved strategies [15,16,17,18,19]. In [15], the infrared image is divided into several sub blocks for detection, which improves the detection speed and robustness. [16] proposed a two-level infrared target detection system combining LCM and local self-similarity, [17] proposed a new local contrast calculation method. [18] proposed a multi-scale patch-based contrast measurement Method (MPCM) and proposed an improved method for high-frequency signal processing in [19]. In order to solve the problem that the aforementioned algorithm cannot adapt to target detection of different sizes, an IR small dim target detection algorithm utilizing the multiscale relative local contrast measure (RLCM) is proposed in [20]. Reference [21] proposed a new ratio difference joint local contrast measure (RDLCM) is proposed between raw IR image and the MDTDLMS result to enhance true small target and suppress all the types of complex backgrounds simultaneously. In [22], a novel local contrast descriptor (NLCD) based on the RW algorithm is proposed to achieve clutter suppression and target enhancement, which enhances the accuracy of target detection. In addition to the method based on local contrast, there are also some articles that analyze other characteristics of the target to detect the target [23,24,25]. In [23], the method of using wavelet transform to analyze the texture direction of the target and calculate the correlation (WTO) to detect the infrared target on the sea surface is proposed, and good results are obtained. Similarly, [24] proposes a direction detection method based on stationary wavelet transform, and determines the region scheme and detection target through self-cross validation. In [25], starting from the feature points of the target, the gray level and gradient of the small target are characterized to analyze the significance of the local properties to detect the target (LIGP), and good detection results are obtained. The method based on HVS generally describes the significance from the gray value and contrast of the neighborhood of a pixel. The algorithm has good reliability, but the effect of eliminating strong interference is not satisfactory. When applied to the infrared image in complex background, it will lead to a large number of false alarms.

Different from the above two kinds of algorithms, there are also some researches on infrared small target detection from the perspective of image data structure. The idea of this method is to extend the traditional infrared model to the infrared block image (IPI) model, and then use the nonlocal autocorrelation feature of the infrared background image and the sparse feature of the small target to transform the small target detection problem into the gray recovery problem of the low rank matrix and sparse matrix, and realize the target detection accordingly. A state-of-the-art approach based on robust principal component analysis (RPCA) called the infrared patch image (IPI) model was proposed in [26]. Then in [27,28,29,30], the algorithm is improved in terms of weight setting and matrix restoration. In [31], a small target detection method base on an image patch sensor (IPT) model is proposed to further optimize the separation process of high-frequency components and low-frequency components. It is worth mentioning that [32] proposes an infrared target detection algorithm (NCL) based on non-convex rank approximation minimization joint l norm, which has achieved good results in the face of complex environment and fuzzy background. The method based on data structure can well adapt to the image with low SNR and complex background environment, but the calculation amount of the algorithm is very large, which is much higher than that based on HVS and traditional algorithm, which makes the practicability of this kind of algorithm very poor.

These methods have achieved fruitful results in the detection of infrared small and weak targets. However, when they are applied to the detection of infrared small dim targets in the maritime background, the performance of the algorithm is limited due to the complex background and strong interference of the infrared maritime image. We choose the following three algorithms to process the real maritime image: WTO which is a more advanced target detection algorithm for the maritime environment; NCL, which is one of the best algorithms for the detection effect based on the image data structure; MPCM, which is based on human visual model, is also one of the most commonly compared algorithms in this research direction.

In Figure 1, the targets in the upper image are two swimmers, while in the lower image the target is a ship. Their characteristics are that they have unique heat radiation characteristics compared to the background, but they only occupy a small number of pixels in the image, so they do not have obvious shape characteristics. In actual maritime search and rescue, people who fall into the water and ships that are far away from the search system show these characteristics. Combined with the above processing results in Figure 1, the following three problems are found:(1)These methods lack of in-depth analysis of infrared image features in the maritime environment, and the algorithm is not effective when applied in the maritime environment.(2)These methods cannot effectively reduce the impact of high brightness island area on the detection process, that is, the effect of background removal is not good, and the edge of the island is easy to be retained to become a false alarm.(3)These methods are not suitable for the characteristics of small and weak targets on the sea, and the effect of target extraction is poor. It is difficult to distinguish them from the strong interference caused by strong waves, which may lead to the weak targets being missed or too many waves being retained.

In order to improve these three problems, this paper proposes the following solutions:(1)In order to solve the first problem, based on the analysis of the common background features of the sea and the features of the infrared dim small targets, this paper proposes a solution of using the image spatial filtering and HVS synthetically, which has a good effect when applied to the maritime environment.(2)In order to solve the second problem, this paper designs a maritime background filter for background removal, which can remove the interference of high brightness island area well while keeping the target area as much as possible.(3)In order to solve the third problem, this paper proposes a small dim target detection operator to extract the target. Through the proposed gradient-magnitude factor and gradient direction deviation two indexes to analyze each region in the image and use the adaptive threshold to segment, the problem that the target is difficult to distinguish from the interference caused by strong waves and island edges is solved, and the false alarm rate is reduced.

In the second section, the features of maritime target and background are analyzed. In the third section, the algorithm flow and principle are introduced. In the fourth section, simulation experiments are carried out and compared with four advanced algorithms. The fifth section expounds the final conclusion.

## 2. Analysis of Maritime Infrared Image Features

### 2.1. Analysis of Background Interference Features

The background interference included in the infrared maritime image is mainly composed of island and wave. In infrared images, the intensity of an object mainly depends on its temperature and radiated heat and it is not influenced by light conditions and object surface features [33]. Next, we will analyze the features of these two kinds of interference. Figure 2 is an infrared image of the sea surface containing small and weak targets, including large-area Island interference. The part of the green box is the island area and its corresponding gray distribution, and the part of the red box is the target area and its gray distribution. From Figure 2, we can see that compared with the target area, the island area not only has an obvious contour, but also has a much higher overall gray value, that is, the gray significance of background interference is stronger, which is not good for our detection.

Different from the island interference, the wave interference does not have obvious contour characteristics, and the distribution has certain randomness. As shown in Figure 3, this is an infrared image of the sea surface containing a small and weak target (marked with a red box). There are obvious waves in the image. It is observed that most of the waves are distributed horizontally and the brightness is far lower than the target, but there are also a few highlighted wave areas which are relatively independent and similar to the target. We use the blue box to select three wave areas that are similar to the target, and analyze the pixel gray distribution of the wave and the target area in the row and column of their respective center points. It is found that in the horizontal and vertical directions, the gray is the local maximum. This is shown in Figure 3.

We find that it is difficult to distinguish the target and the wave from the gray features, but the difference between the gray of the target area and the gray of the adjacent area is greater than that of the wave area. So we extract the gradient features of the image, as shown in Figure 4, which are the gradient amplitude of each region in the original image, the horizontal gradient value and the vertical gradient value.

It can be seen from the above Figure 4 that the target area has high gradient values in both horizontal and vertical directions, while most of the waves have high gradient values only in vertical direction. Although there is a small difference between the gradient amplitude and the target area, the gradient value of the three marked wave areas similar to the target area is still much smaller than the target area in the horizontal direction.

Generally speaking, the two main interferences in the infrared maritime image have different characteristics. The island interference has high brightness and obvious contour. However, the wave interference does not have obvious contour features, but the gray level in the local area is more prominent, which is similar to the target to some extent, and the gradient features are quite different from the target area. This provides a theoretical basis for the design of maritime background filter, but in order to better retain the target area when removing the background, we need to further analyze the features of the target in the infrared maritime image.

### 2.2. Analysis of Target Interference Features

#### 2.2.1. Analysis of Target Gray Scale Features

Four infrared maritime images including small dim targets are listed in Figure 5. It can be found that small and weak targets do not have any contour and texture features, and occupy very few pixels, ranging from a few pixels to a dozen pixels. Not only that, the brightness of each target will also vary with its own radiation intensity.

According to the above analysis, morphological features, such as contour and texture, cannot be used in the detection of small and weak targets. To distinguish the target from the background, we can only start with the gray-scale features. Next, we will analyze the gray-scale features of the target.

The gray distribution map of each image shown in Figure 5. It can be seen from the figure that the gray value of the target area is usually the high point of the whole image. Even if there is a large number of strong wave interference with high gray value in the second image, the target area still has a high significance in the gray characteristics, which can be used as the basis for us to detect the target.

At the same time, we realize that when there are more strong wave interferences in the image, a large number of interferences will be retained when the target is detected only based on the gray value, which we do not want to see. It can be seen in Figure 3 that although the gray values of the target area and the wave area are relatively prominent, the gray differences between the target area and the adjacent pixels in the horizontal and vertical directions are larger, that is, the gray gradient values in the two directions are larger. That is to say, the high gradient value in both horizontal and vertical directions is the unique feature of the target area, which can help us distinguish the interference caused by the target and strong waves effectively.

#### 2.2.2. Similarity between Target and Salt Noise

Through the observation of a large number of infrared maritime images containing small and dim targets, we found that these small and dim targets have the morphological characteristics close to salt and pepper noise. Pepper and salt noise, also known as pulse noise, is a kind of noise often seen in the image, pepper represents black, salt represents white. Because in the infrared maritime image, the gray value of the target is higher than that of the background, that is to say, it is brighter. So the target is closer to the “salt noise” in salt and pepper noise.

The area marked with red box in Figure 6 contains a small target. Next, we introduce extremely low density salt noise into the image to observe the similarity between noise and target.

In Figure 6, the target is marked with a red box, and four random salt noises are marked with a blue box. In order to compare the local gray characteristics between the two, we draw the gray distribution of 15 × 15 pixels near each region. It can be seen that the gray values of the target and the noise area are at the highest point in the local range. This shows that the salt noise has a very high similarity with the small and weak targets on the sea surface.

This unique similarity can provide ideas for the design of maritime background filter. As mentioned above, the design of background filter should not only consider the characteristics of the background, but also analyze the characteristics of the target to better retain the target area. In the field of image processing, median filtering is the most commonly used method to remove the salt noise. Because of the similarity between the target and the salt noise, we can use the idea of median filtering to design the maritime background filter. After processing the image, the target will be removed, and the remaining part is the background area. The target area can be extracted by the difference between the background image and the original image. However, the traditional median filter can’t retain the edge details of the image, which makes the edge of the island in the difference image also be preserved, affecting the subsequent detection, so we consider this problem and make improvements in the design of the maritime background filter.

In addition, salt noise is a random white spot in the image, which has no correlation with the background. It has a large difference with the gray level of surrounding pixels, and has a high and close gradient value in the horizontal and vertical directions. Similarly, the appearance of the target in the real scene is also random, and has the characteristics of thermal radiation different from the sea surface, and the gray level of the surrounding pixels is also very different, which again shows that the gradient value of the target in the horizontal and vertical directions is high and close. This provides a theoretical basis for the design of target extraction operators.

## 3. Target Detection Algorithm Based on Maritime Background Filtering and Target Extraction

Based on the analysis of maritime targets and background features, the framework of the target detection algorithm proposed in this paper is shown in the figure. The method is mainly divided into two steps: background eliminating and target extraction. The first stage: based on the analysis of maritime background features in Section 2.1 and the similarity between the target and salt noise in Section 2.2.2, a kind of maritime background filter based on the median filter of sensitive region selection is designed. The filter can remove the background well on the basis of accurately estimating the complex background of the image combined with the image difference technology. The second stage: according to the features of the target analyzed in Section 2.2, a detection operator of small dim targets based on the gradient feature description is proposed. Through this operator, the difference between the target and the interference can be expressed and the target can be accurately extracted by combining the adaptive threshold. In order to make the result more obvious, the resulting image is subjected to a morphological expansion operation to obtain the final target detection result. Flow chart of the proposed method is shown in Figure 7.

At the same time, the flow chart of the algorithm (SRGM) proposed in this paper shows the key steps in Figure 8. Combining Figure 7 with Figure 8, it will be easy to understand the process of our algorithm more clearly.

### 3.1. Design of Maritime Background Filter

The maritime background filtering needs to extract the background of the image first, and then differentiate the extracted image and the original image to get the suspected target area. The design of the maritime background filter needs to consider both the background and target features of the image. For the infrared maritime image, the background is mainly a large area of slowly changing low-frequency part, with less details. However, the small dim targets show a high similarity with the salt noise.

In order to achieve accurate background extraction, we propose a median filter based on sensitive area acquisition, which is mainly divided into two parts: sensitive area acquisition and sub area processing, as shown in Figure 9.

Considering that the median filter has a good performance in dealing with salt noise, and the processing process has little change to the smooth background. So we use median filter to extract image background. The standard median filter will process every pixel in the original image, so the image processing results will be affected by the set filter window size, and it cannot retain the edge information in the image, such as the island edge will be largely blurred. The standard median filter can be expressed by Equation (1), f(x,y) is the gray value of the pixel in the original image at coordinates (x,y), b(x,y) is the gray value of the pixel in coordinates (x,y) after the median filter, and R is the calculated local area:(1)b(x,y)=median[f(x,y)]         x,y∈R

The proposed filter can solve these two problems when the standard median filter is used to extract the background. First, the original image is filtered by a large window median filter (7 × 7 in this paper when the size of original image is 320 × 256) to get the rough estimation image of the background. Then the image is differentiated from the original image, and the difference result image can represent the gray level change degree of each pixel in the original image after median filtering. The larger the change is, the higher the gray value is in the difference image, which is the sensitive area defined by us. For the most sensitive target area pixels of median filter, the difference image shows high gray level, while the island area pixels show low gray level. In order to distinguish the target and background better, we use gamma transform with gamma coefficient of 2 to get the stretched image with larger gray-scale difference. We calculate the gray-scale mean *T* of all non-zero pixels in the stretched image as the threshold value. The calculation formula of *T* is shown in (2). *M* is the set of all non-zero pixels in the stretched image, and $m_{n}$ represents the gray value of the nth pixel:(2)$$T=\frac{\sum^{\;}M}{n},\;M=\left\{m_{1},m_{2},m_{3},m_{4}\ldots m_{n}\right\}$$

The original image corresponds to the pixel area in the stretched image where the gray-scale is higher than the threshold value *T*, which is considered as the sensitive area, and the area lower than *T* is considered as the non-sensitive area domain.

For the pixels in the sensitive area, a small window median filter will be carried out (3 × 3 is selected in this paper), while for the pixels in the non-sensitive area, we will keep their gray values unchanged. In this way, the number of pixels processed by the filter window is reduced, and the influence of the filter window size on the processing effect is also reduced. At the same time, only the sensitive areas of similar targets are filtered, while the large area of the background area is well preserved. In this way, the accurate image of maritime background is obtained.

After obtaining the accurate extraction image of the maritime background, the difference between the image and the original image can eliminate most of the background, and the simple gray threshold segmentation of the result image can further eliminate the clutter interference and more accurately screen out the suspected target area. The results of extracting target area from four images after standard median filter and improved median filter are shown in Figure 10.

It can be seen from the above figure that the maritime background filter designed in this paper has achieved good results in background removal, and the interference of typical waves and island edges has been well suppressed, and the suspected target area can be accurately extracted for further judgment and screening.

### 3.2. Design of Small Dim Target Extraction Operator

Based on the gradient features of the pixels in each suspected region, the small dim target extraction operator proposed in this paper will analyze and judge each region to further eliminate the interference and extract the target. According to the analysis in Section 2.2, the target area has the following two features: large gradient amplitude and similar component size of gradient in horizontal and vertical directions. The second characteristic can also be expressed as the gradient direction is close to 45° after transforming to the first quadrant. In order to ensure that the target with low contrast can be detected and the strong wave interference can be eliminated as much as possible, the detection operator designed by us includes two analysis indexes: gradient-magnitude factor and gradient direction deviation. In order to accurately extract the target, we will use these two operators to judge each suspected target area in turn. Finally, the morphological expansion of the detection results is carried out to make it more obvious.

#### 3.2.1. Gradient-Magnitude Factor

The traditional definition of gradient magnitude can be described by equations (3) and (4), in which the gradient value of a pixel in the horizontal direction is $G_{X}$, the gradient value in the vertical direction is $G_{Y}$, and the gradient magnitude is $G_{L}$; $G_{x}$ and $G_{y}$ are Sobel operators in the horizontal and vertical directions respectively, and *Img* is the image region involved in the calculation:(3)$$G_{X}=G_{x}*Img\;G_{Y}=G_{y}*Img\;,G_{x}=\left[\begin{array}{ccc} -1 & 0 & +1\\ -2 & 0 & +2\\ -1 & 0 & +1 \end{array}\right]G_{y}=\left[\begin{array}{ccc} +1 & +2 & +1\\ 0 & 0 & 0\\ -1 & -2 & -1 \end{array}\right]$$
(4)$$G_{L}=\sqrt{{G_{X}}^{2}+{G_{Y}}^{2}}$$

It is found that the traditional gradient magnitude cannot describe the features of the target very well. The target has a considerable gradient value in the horizontal and vertical directions, while the strong waves usually only have a high gradient value in a single direction. In this way of analysis, the gradient value in one direction contributes a lot to the index, that is to say, a strong interference may be caused by a high gradient value in one direction However, the strong interference caused by some strong waves is more prominent than the target with low contrast. In order to solve this problem, we propose a new gradient-magnitude factor $G_{V}$, and it is obtained by multiplying $|G_{X}|$ and $|G_{Y}|$. Compared with the traditional index, the contribution of high gradient value in one direction to the result is restrained, and the more balanced pixels in two directions will be more prominent. This index can be expressed by Equation (5):(5)$$G_{V}=|G_{X}|\times |G_{Y}|$$

The advantage of the proposed gradient-magnitude factor compared with the traditional gradient magnitude is that for those cases that cannot be distinguished when using gradient magnitude for analysis, our proposed index can be well differentiated. Assuming that $G_{L}$ is certain, there are countless combinations of $G_{X}$ and $G_{Y}$ values, and each combination may be a gradient feature of interference region. In this case, it is impossible to distinguish interference and target by index $G_{L}$. The relationship between $G_{V}$ and $G_{X}$ is described by Equation (6):(6)$$G_{V}=|G_{X}|\times \sqrt{{G_{L}}^{2}-{G_{X}}^{2}}\;|G_{X}|\in \left[0,L\right]$$

When $G_{X}$ = $G_{Y}$, $G_{V}$ gets the maximum value. According to the previous analysis, the gradient feature of the target area is the closest to $G_{X}$ = $G_{Y}$. That is to say, when $G_{V}$ is selected as the analysis index, the target area will be in the most prominent position among many candidate areas. 

We use gradient magnitude and gradient-magnitude factor to describe the gradient features of the image. The results are shown in Figure 11. It is difficult to distinguish the target from the wave or the island edge when we use the gradient-magnitude factor. However, when we use the gradient-magnitude factor proposed by us, the difference between the target and the interference is very obvious, and the significance of the target is much higher than all kinds of interference. In this case, the gradient-magnitude factor $G_{V}$ proposed by us is more suitable for target extraction.

After quantizing the pixels of each suspected area by the gradient-magnitude factor, the image can be adaptively segmented using (7) extract the target:(7)Thf=avg+m×std

In the formula, *Thf* is the established segmentation threshold, avg is the average value of the analysis indexes calculated for all the suspected areas, std is the standard deviation, and *m* is an adjustable weighting factor (0.78 in this paper). The part above the threshold will be considered as the real target area and retained, and the gray value of the remaining part area will be set to zero to eliminate.

#### 3.2.2. Gradient Direction Deviation

In order to further extract the target accurately, we will use the gradient direction deviation to describe the similarity of the gradient in the two directions to further judge each suspected target area. The gradient direction deviation $N_{d}$ is an index only related to the similarity of gradient value in two directions, but not related to gradient value. The closer the gradient value of a pixel in two directions is, the closer the gradient direction is to 45 degrees. So we can use the degree of difference between gradient direction and 45 degrees to indicate the degree of difference in gradient value in two directions. In this case, $N_{d}$ can be used to further eliminate the suspected target areas that do not meet the target features. It can be described by Equation (8):(8)$$N_{d}=\left|\arctan\frac{\left|G_{Y}\right|}{\left|G_{X}\right|}\;-\;45\text{°}\right|$$

In the same way, after quantifying each region with the gradient direction deviation index, (7) is used to segment the image adaptively, which can further eliminate the interference and extract the target accurately.

#### 3.2.3. Morphological Expansion of Image

The features of small dim target determine that the target area only occupies a very small number of pixels. In order to make the target with a small number of pixels in the detection results easier to be observed, morphological dilation operation will be performed on the detection results. It can be described by Equation (9):(9)$$A⊕B=\left\{z\;|\;{\left(\hat{B}\right)}_{z\;}⋂\;A\neq \emptyset \right\}$$

Morphological expansion of image refers to the convolution of some images (or a part of an area in an image, called A) with a kernel (called B). The kernel can be viewed as a template or mask, and dilation is an operation to find a local maximum. Kernel B is convolved with the image, that is, the pixel maximum value of the area covered by kernel B is calculated, and this maximum value is assigned to the pixels specified by the reference point, so that the highlighted area in the image will gradually increase. We use a circular convolution kernel with a diameter of three pixels to perform the dilation operation on the detection results. 

In this section, based on the analysis of target and background interference features in Section 2.2, a detection operator for small dim targets based on gradient features is proposed, which includes two analysis indexes: gradient-magnitude factor and gradient direction deviation. By analyzing and judging the pixels in each suspected area in turn, the real target can be accurately extracted from the suspected target area extracted in the previous section. All kinds of background interference will be eliminated to the greatest extent.

## 4. Experimental Data and Result Analysis

In order to verify the feasibility and reliability of the algorithm in many cases, in this section, firstly, several typical infrared maritime images including small dim targets are selected and analyzed, and the original contrast features of the targets are analyzed. Then our algorithm (SRGM) is used to process the single frame image, and the detection results are analyzed. At the same time, the detection results of our algorithm (SRGM) are compared with four advanced infrared small target detection algorithms.

It should be noted that the computer platform parameters during our experiments are as follows: CPU: Intel i7-8750H @2.20 GHz, GPU: NVIDIA GeForce GTX1060. When the algorithm proposed in this paper runs in this environment (Microsoft Visual Studio 2013& Opencv2.4.9), for images of size 320 × 256, the detection speed is 31.23 fps. And for images of size 640 × 512, the detection speed is 12.2 4 fps.

### 4.1. Experimental Data

Seven kinds of infrared maritime images including small dim targets are shown in Figure 12. Each image is taken by a mid-infrared cooled thermal imager with a response frequency of 3.7–4.8 μm. the pixel is 320 × 256, and the output frequency of the thermal imager is 25 fps.

All the real small dim targets in Figure 13 have been marked with boxes. In this paper, these images will be used to test the performance of the algorithm. Equation (10) is used to calculate the gray contrast of the target in the figure:(10)$$\mathrm{contrast}=\frac{\bar{g_{t}}-\bar{g_{b}}}{\bar{g_{b}}}$$

In Equation (10), $\bar{g_{t}}$ represents the average gray value of the pixels in the target area, $\bar{g_{b}}$ represents the average gray value of the background pixels in the target neighborhood, and contrast represents the gray contrast of the target area.

According to the image in Figure 12 and the data in Table 1, the background and target features in the above seven images have their own characteristics. In Figure 12a, the background is simple and the sea surface is calm, but there is a weak small target A2 with very low gray contrast. In Figure 12b, the brightness of sea surface is uneven, and there are three small targets with relatively high gray contrast. In Figure 12c, the background also includes the sea surface and the sky. There are a lot of strong wave interference, and the difference between the target and the background is small. In Figure 12d, the background includes the sea and sky, and the brightness of the sea and sky are quite different. The two targets are located on the horizon line in the field of view. In Figure 12e the background is complex, including large and bright islands, obvious horizon line, clouds and waves, and the target brightness is low. There is serious sea fog in Figure 12f, and the brightness distribution is very uneven. In Figure 12g, the background is complex, and the target is located on the right side of the ship, accounting for only 1–2 pixels. The contrast is very low, so it is a very weak target. According to the above analysis, the six selected images not only meet the ideal conditions of image processing, but also have images that are difficult to detect due to various objective reasons, which include most of the situations that can be encountered in the actual maritime search task. To a certain extent, using these image data can verify the performance and practical value of the algorithm designed in this paper.

According to the above analysis, the selected seven images are both those that meet the ideal conditions for image processing and those that are difficult to detect due to various objective reasons, and these objective reasons include most of the situations that can be encountered in actual maritime search tasks. Therefore, to a certain extent, using these image data can well verify the performance and practical value of the infrared maritime small target detection algorithm designed in this paper.

### 4.2. Single Frame Detection Result Image and Analysis

This part shows the detection results and analysis of the algorithm for single frame image. In this paper, four more advanced infrared image single frame detection algorithms are compared with our algorithm (SRGM). They are infrared maritime target detection algorithm (WTO) based on texture direction feature, infrared target detection algorithm (NCL) based on non-convex rank approximate minimization combined l norm, infrared small target detection algorithm (MPCM) based on multi-scale patch contrast measurement and Small infrared target detection algorithm (LIGP) based on local intensity and gradient performance. The test results are analyzed and the test performance is summarized. Through the detection of seven typical infrared maritime images under different backgrounds in Section 4.1, the detection results and performance are analyzed and summarized.

The detection results of the five algorithms are shown in Figure 13. The five algorithms are used to detect eight typical infrared maritime images with different background complexity. For Figure 13a,b with smooth background, there are many targets with different contrast. Except that all targets are successfully detected by our algorithm (SRGM), other algorithms miss the detection of low contrast targets to varying degrees. However, all algorithms do not retain background interference.

For Figure 13c with a large number of strong wave interference in the background, all algorithms can detect the target. At the same time, there is no residual background interference in our, WTO, NCL, while there is a certain wave interference in the detection results of MPCM and LIGP.

For Figure 13d of the target on the horizon line, all algorithms can successfully detect two targets and filter the background interference.

For Figure 13e with island interference, SRGM successfully detects the target and removes the background interference. WTO, MPCM, LIGP and NCL all detected the target, but they all kept the edge of the island to varying degrees.

For Figure 13f with dense fog interference, although the gray difference between the upper and lower parts of the image is large, the background is relatively smooth, all algorithms can detect the target, and the background filtering effect is very good.

For Figure 13g with very weak target, only the SRGM can detect the target successfully.

Through the above experimental results, we found that when the background is relatively smooth, the five algorithms have good filtering effect on the background. When there are strong waves, islands and bridges interference, WTO, MPCM and LIGP are easy to introduce interference, and the background filtering effect of our and NCL is relatively stable. In terms of target characteristics, when the target gray value is high and the local contrast is high, the five algorithms can easily detect the target. However, when the gray value of the target is very low, the local contrast is very low, and the background is very close or the number of pixels occupied is too small, it is difficult for WTO, MPCM, NCL and LIGP to successfully detect the target, and SRGM can still accurately detect the target, which shows that the algorithm proposed in this paper (SRGM) has a good performance on the extremely small target with low gray value and few pixels occupied.

### 4.3. Sequence Image Detection Result and Analysis

According to the different background features, the above images can be summarized into five categories: Island interference Figure 14a, target on the sea sky line Figure 14b, calm sea surface Figure 14c, dense fog sea surface Figure 14d and strong wave interference Figure 14e. These five kinds of images cover most of the common sea environment. We will use these five kinds of image sequences to quantitatively evaluate the performance of the algorithm. The first four types of image sequences contain 100 images. Due to the limitation of image database, only 75 images are included in the image sequences under strong wave interference.

We will select MAR (Missing Alarm Rate) and FAR (False Alarm Rate) as two evaluation indexes, which are defined as (11) and (12). In (11), MT refers to the number of missed targets, DT refers to the number of detected targets, and in (12), FD refers to the number of incorrectly detected targets, that is, the number of targets involved in interference detection:(11)$$\mathrm{MAR}=\frac{\mathrm{MT}}{\mathrm{DT}+\mathrm{MT}}\times 100\%$$
(12)$$\mathrm{FAR}=\frac{\mathrm{FD}}{\mathrm{DT}+\mathrm{FD}}\times 100\%$$

MAR of five detection algorithms is shown in Table 2. FAR of five detection algorithms is shown in Table 3.

As far as MAR is concerned, our algorithm (SRGM) and NCL perform best, and there is no missing detection in 475 detected images. The WTO has about 9% missed detection rate in sequence A and sequence B. According to the image, the target gray level in sequence a is low, and the wave in the background is strong, which causes strong interference to the detection. The three methods of WTO, MPCM and LIGP also have some omissions in the detection of sequence B, which is caused by the target appearing on the horizon line, the strong wave interference and the low background contrast of some targets.

As far as FAR is concerned, in addition to sequence D, our algorithm (SRGM) shows the best performance among the five algorithms, and with the simplification of the background, the FAR value of the algorithm in this paper will be lower and lower, which means that for the vast majority of cases, the algorithm proposed in this paper will introduce the least interference in the detection. But in sequence D, the FAR of the algorithm proposed in this paper is only 4.15% higher than that of MPCM with the best performance. However, combining with Table 3, it is found that MPCM has a slightly lower FAR and a very high MAR, which is not desired in practical application.

According to the above experimental results and data, it is shown that after fully analyzing the maritime background and target features, the proposed detection method (SRGM) finally obtains stable and excellent detection results in various maritime environments, and has a certain timeliness, with high application value.

## 5. Conclusions

In order to propose a single frame detection algorithm for small dim targets in infrared maritime images, this paper first analyzes the characteristics of the common interference on the sea and the weak small target, and proposes a detection algorithm (SRGM) based on filtering the maritime background and the accurate extraction of the target. The algorithm first processes the image through the designed maritime background filter, obtains the accurate estimation of the background area, then uses the image difference to screen out the suspected targets which conforms to the features of the small dim target, then uses the proposed small dim target extraction operator to carry on the quantitative analysis to each suspected area, and combines the adaptive threshold to judge each suspected target area and finally retains the qualified target area as the result.

Finally, it is verified by experiments that the algorithm (SRGM) proposed in this paper has high accuracy and real-time performance. It can detect all the targets when it is used in various typical sea scenes. That is, the missing alarm rate of our algorithm is zero which will be very beneficial for practical application. At the same time, our false alarm rate is also the lowest among the algorithms, which means that our algorithm will retain less interference. Not only that, the detection speed of our algorithm can achieve 31.23 fps on average for images of size 320 × 256., and 12.24 fps for images of size 640 × 512. Generally, our algorithm (SRGM) can accurately and quickly detect the existence of small dim targets in a variety of maritime backgrounds, and has high practical value.

## Figures and Tables

**Figure 1 sensors-20-01237-f001:**
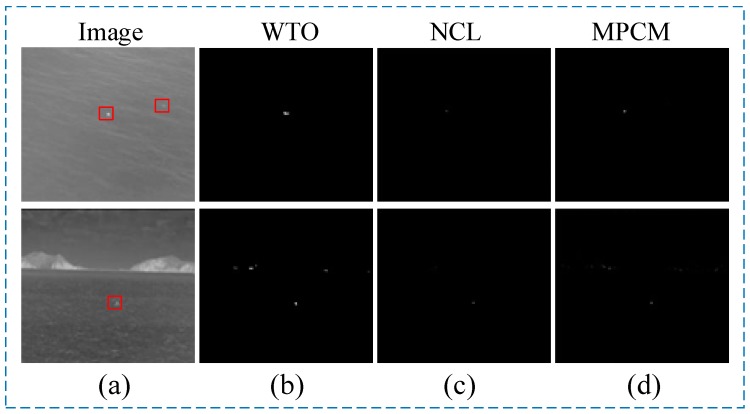
Results of three target detection algorithms in two typical maritime background: (**a**) denotes the original images. For the upper image in (**b**–**d**), they have only detected the target on the left. For the lower image in (**b**–**d**), they have detected the target but kept island and wave interference.

**Figure 2 sensors-20-01237-f002:**
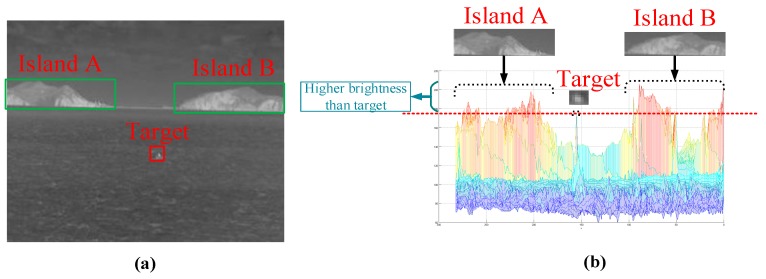
Infrared maritime image including island and target. (**a**) denotes the original image, (**b**) denotes the gray distribution of each area.

**Figure 3 sensors-20-01237-f003:**
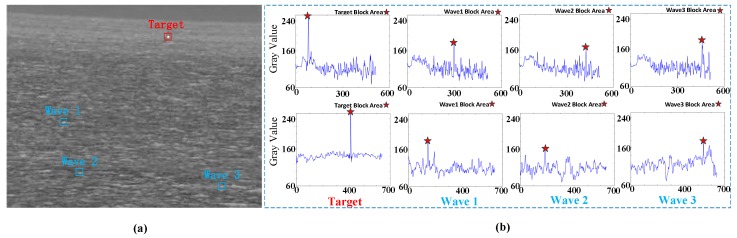
Gray distribution of waves and target. (**a**) denotes the original image (640 × 512), (**b**) denotes the gray distribution of waves and target in two directions, the upper part is the vertical directions and the lower part is the horizontal directions.

**Figure 4 sensors-20-01237-f004:**
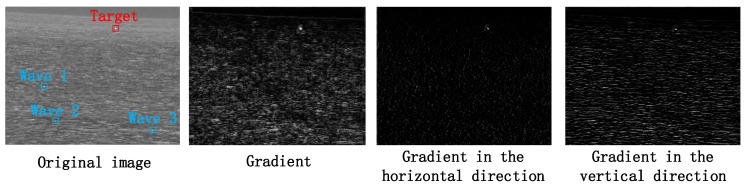
Gradient distribution of waves and targets.

**Figure 5 sensors-20-01237-f005:**
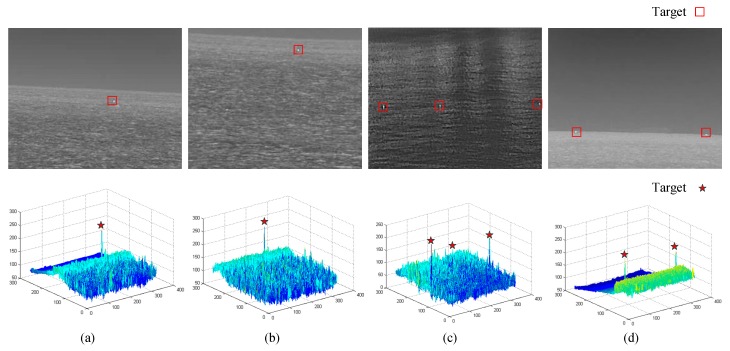
Infrared maritime images including small dim targets: The upper parts of (**a**–**d**) are the original images and the lower parts are the gray distribution maps of each image.

**Figure 6 sensors-20-01237-f006:**
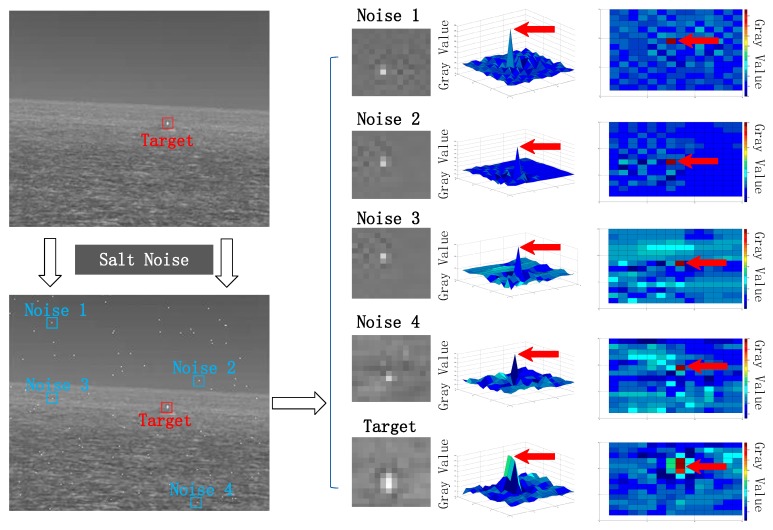
Gray distribution of target and salt noise.

**Figure 7 sensors-20-01237-f007:**
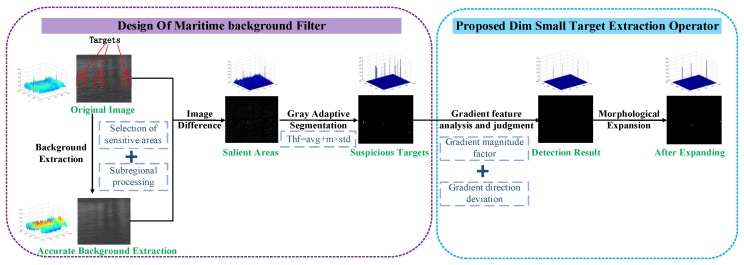
Flow chart of the proposed method.

**Figure 8 sensors-20-01237-f008:**
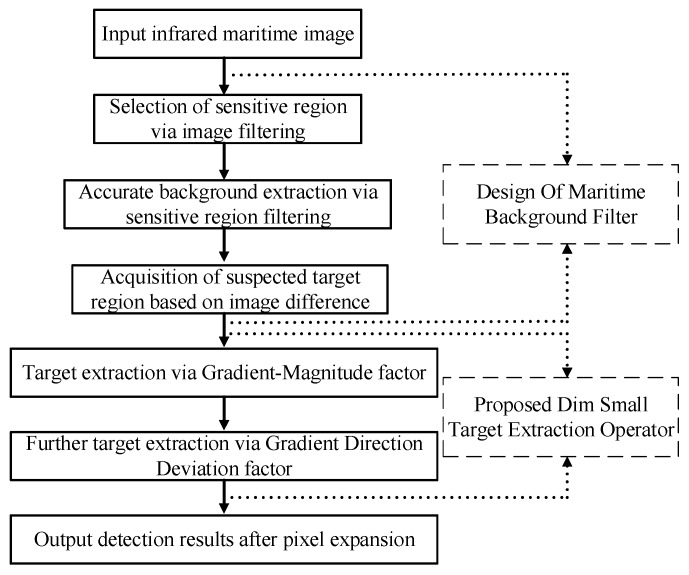
Flow chart of the proposed algorithm (SRGM).

**Figure 9 sensors-20-01237-f009:**
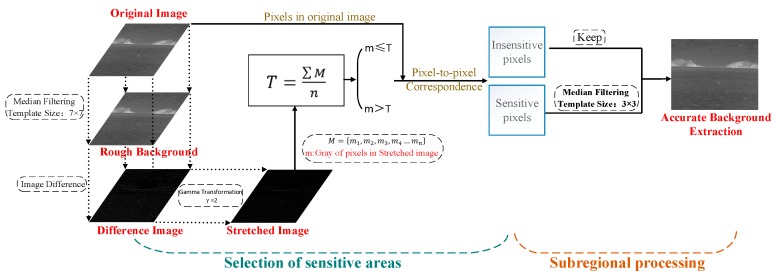
Flow chart of the proposed median filter based on sensitive area acquisition.

**Figure 10 sensors-20-01237-f010:**
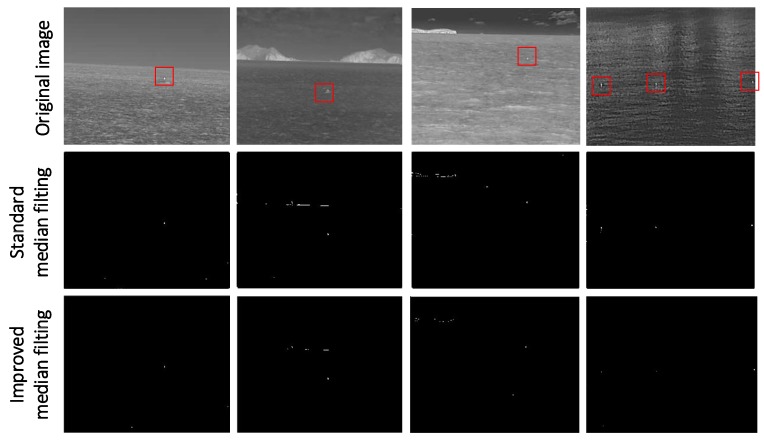
Comparison of the effect of two kinds of filters for target extraction.

**Figure 11 sensors-20-01237-f011:**
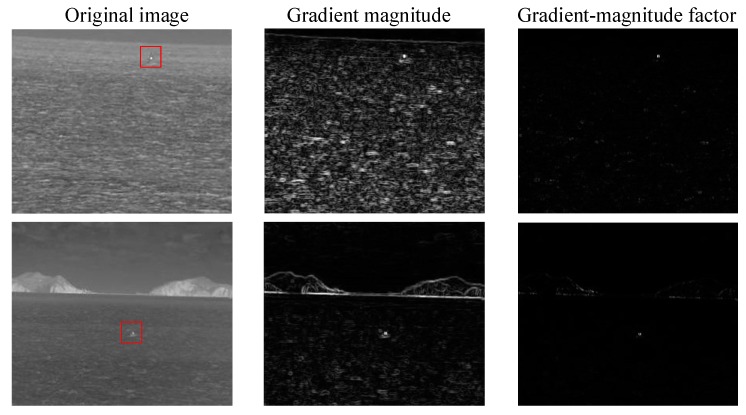
Result of using $G_{L}$ and $G_{V}$ to analyze gradient in turn.

**Figure 12 sensors-20-01237-f012:**
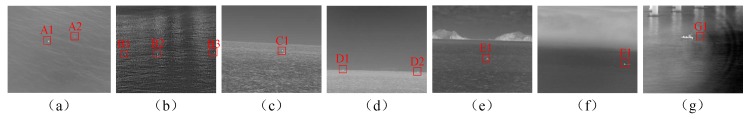
Experimental image:(**a**) two targets with different contrast on a calm sea; (**b**) three targets with different contrast on the sea with uneven brightness; (**c**) target on the sea with obvious waves; (**d**) two targets on the horizon line; (**e**) island interference in the image; (**f**) fog interference in the image; (**g**) extremely small and dim target.

**Figure 13 sensors-20-01237-f013:**
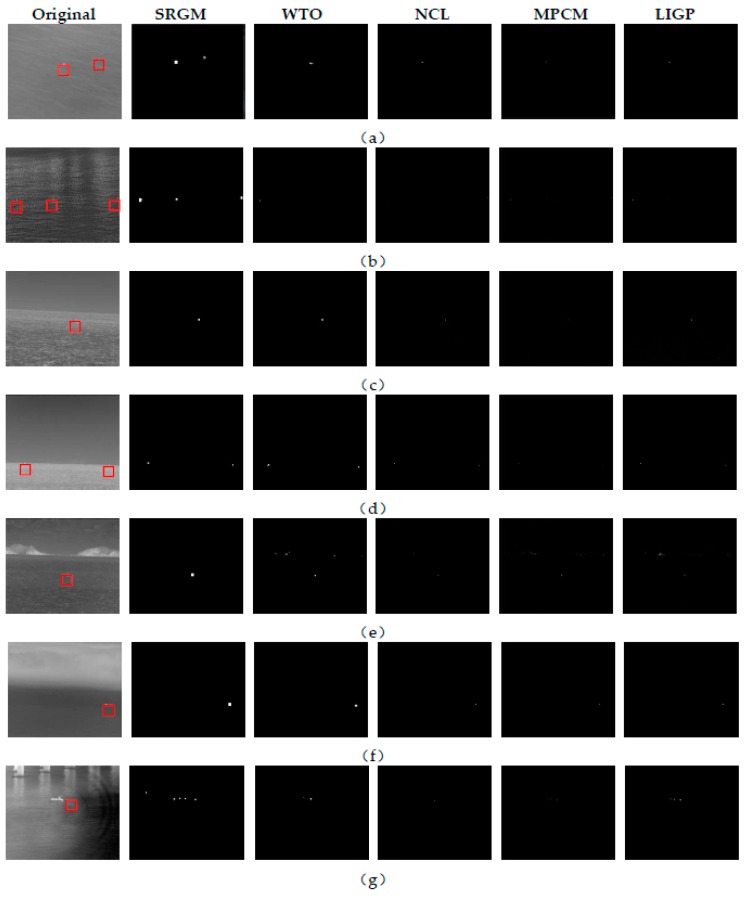
Detection results of the above five algorithms: (**a**) two targets with different contrast on a calm sea; (**b**) three targets with different contrast on the sea with uneven brightness; (**c**) target on the sea with obvious waves; (**d**) two targets on the horizon line; (**e**) island interference in the image; (**f**) fog interference in the image; (**g**) extremely small and dim target.

**Figure 14 sensors-20-01237-f014:**
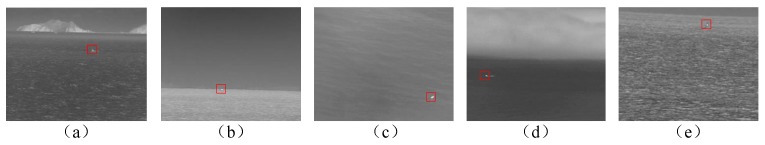
Five types of typical maritime images: (**a**) island interference in the image; (**b**) target on the horizon line; (**c**) target on the calm sea; (**d**) fog interference in the image; (**e**) target on the sea with obvious waves.

**Table 1 sensors-20-01237-t001:** Contrasts of infrared Maritime Targets in Figure 12.

Target	A1	A2	B1	B2	B3	C1	D1	D2	E1	F1	G1
Contrast	0.23	0.04	0.23	0.40	0.45	0.12	0.23	0.14	0.33	0.73	0.01

**Table 2 sensors-20-01237-t002:** MAR of five detection algorithms.

Image	Number	MAR(%)				
		SRGM	WTO	NCL	MPCM	LIGP
A	100	0	9.09	0	0	0
B	100	0	9.09	0	4.55	4.55
C	100	0	0	0	4.35	0
D	100	0	0	0	0	0
E	75	0	0	0	0	0
Total	475	0	3.88	0	1.94	0.97

**Table 3 sensors-20-01237-t003:** FAR of five detection algorithms.

Image	Number	FAR(%)				
		SRGM	WTO	NCL	MPCM	LIGP
A	100	64.52	90.48	65.63	81.97	85.71
B	100	12.00	48.72	15.38	12.50	12.50
C	100	0	30.30	0	31.25	11.54
D	100	8.70	34.38	16.00	4.55	8.70
E	75	6.25	44.44	42.31	48.28	58.33
Total	475	30.87	70.97	37.20	53.88	61.62

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
