# Peer review of "A Robust Detection Algorithm for Infrared Maritime Small and Dim Targets"

_sensors, 2020, doi:10.3390/s20041237_

Round 1

Reviewer 1 Report

This is a magnificent work, which clearly provides new knowledge for future research. The ideas that justify the magnificent experimental and comparative results are very interesting.

However, I have detected a multitude of errors and issues that need to be resolved for possible publication:

In the introduction: Check the punctuation marks before and after the references to the bibliography, as there are several "dots" left over or to be changed by commas, at the beginning of the next sentence in lower case.

In line 123 and figure 1, reference is made to the "WT" algorithm, but it is not mentioned in the introduction. I understand that it is the "WTO".

In figure 1, what kind of targets are you trying to detect? What are their characteristics? What is the real practical application of this detection? Are we talking about military targets, animals, other ships? I understand that these are hot spots in the image, but I think it would be useful to better describe what you are looking for since you do not see any shape in the reference images in the figure. Above all, it is difficult to see, even for a human eye, that the marked target on the right of the upper image in figure 1, is something distinguishable from waves.

With respect to the figure caption in figure 1, normally the figure captions illustrate what is seen in the figure in more detail and if the figure is composed of several sub-pictures, they are numbered with letters.

In line 161 I think that figure 2 should be referred to instead of figure 1.

In line 162 we talk about a green box when in figure 2 there is no green box, but all red boxes. I think that figure 2 should be modified to show the green boxes and not the red ones on the islands.

The caption of figure 2 can clearly be improved to illustrate the different subpictures of the figure in more detail.

In line 170, should not figure 3 be referenced?

In figure 3, on the right hand side, are the upper part the horizontal directions and the lower part the vertical ones? I think it should be specified correctly in the figure's caption, as well as following the usual indications of numbering each subpicture with correlative letters and explaining in the caption what each one is.

Also, the numerical values of the pixels are not very well seen. I think this aspect should be improved, as well as the overlapping of texts with different fonts.

In line 182, instead of "in the following figure" I think you should put "in figure 4".

I think that figures 5 and 6 can be put together in one figure, numbering each sub image with correlative letters and explaining in the figure's caption the characteristics of each image.

In the distributions of figure 6, I think it would be clearer if they are rotated so that they correspond better to the reference images, that is, that the sky is seen at the top and not on the right as it is now. And the title of the vertical axis appears in Chinese!

The conclusions of the paragraphs in lines 213 and 218, I think are reasonable, but I think it would be clearer if the study of figures 5 and 6 were done before that of figures 3 and 4. This would justify the problem of level detection of the secure image of figure 5 and the need to study figures 3 and 4 and therefore the use of the gradient.

Line 233 talks about "some noise density", I think it should be better specified what density and its influence on detection.

I don't know if it is really necessary to make the study of figure 8, I think it is evident that the salt and pepper noise is different from the other noises and it is the one that most resembles to the objectives sought in thermal images.

On line 250, the word "More" appears with a capital "M".

In lines 305 to 308, the nomenclature of the variables should be revised to correspond to equation 1 (in italics and lowercase). And that no space is left between the variable and its parameters.

Line 310 talks about a 7x7 filter, I think it is necessary to justify the specific value of 7x7 and to relativize it with respect to the size of the image.

In line 318 and 320, the variables "t" and "T" must be in italics. Are they the same? If not, what is the difference between them? How is T calculated? What is the M and the m in Figure 10. All these aspects should be explained better and the equation in Figure 10 should be put in the text as one more equation.

In line 321, the word "Domain" with a capital "D" appears.

The paragraph in line 333 repeats what has already been explained above. I think you can delete it and leave only the novelty.

In equation 2, the operators of Sobel Gx and Gy, it would not hurt to put them in the text, instead of giving them as known.

In equation 3, the operator "x" I understand is a simple multiplication, right? If this is the case, it is not superfluous to indicate it in line 381 to avoid confusion with the subsequent convolution, for example.

In Figure 12, if possible, it would be better if both GL and Gv were put in italics with the subscript.

The text in lines 404 to 408 of the explanation of equation 5, should go after the equation and not before it. And all variables should be italicized. Is the std the standard deviation or the variance? Because m=0.78. It should be explained how this practical value was arrived at.

In point 3.2.2, I think that the Nd should be explained a little better and especially for what purpose it is subtracted from 45 degrees. In formula 6, please do not leave the space between the vertical bars and the expression.

On line 449, the "M" should be "m" and be next to the microns.

Figures 13 and 14 are different experimental images, so at the bottom of the figure you can indicate the main characteristics of each one as a summary.

On line 463, isn't A2 the one with the low contrast? Figure 13c has no islands... I think we have to check the correspondence between figure 13 and the text.

In figure 13(g) and figure 14(g) I think the target box is shifted to the right and does not mark the actual target to be searched.

In lines 511 to 526, the number of the referenced figure is missing, in this case 14.

In line 518, what do you mean by sea antennas in figure 14d?

In lines 521, 530 and 535, "ligp" should be in capital letters like the rest of the algorithms.

Before the paragraph starting with line 539, I think there should be a point 4.3 to distinguish the results with the 7 simple images from the more comprehensive study with the image gallery of the different categories.

In line 539 "be" is separated "b e".

In lines 539 to 541, references to figure 15 are missing.

In line 541, the letter "E" should be in lower case.

The footnote in figure 15 is in a different typeface from the rest.

Equations 10 and 11 use an unusual nomenclature, typically using the usual Sensitivity and Specificity parameters. In the following reference you can see more details: https://en.wikipedia.org/wiki/Sensitivity_and_specificity  

Please change the MAR and FAR formulas and parameters in the tables and text following that nomenclature, which should also always be in capital letters so that they are not confused with the word "far" in English.

The mention of the temporal analysis that starts in line 575 is very confusing, actually it has not been possible to do it, no? If so, perhaps it is better not to talk about this topic and to indicate at the beginning of point 4 the data of table 4.

The proposed algorithm should have a name of its own and not simply be called "our" as this will allow it to be referenced in the future by other authors who use it.

Reviewer 2 Report

The work is good, but still needs some more improvement to make it much better: 

In order to have the work of computer engineering type, an algorithmic description of the given flow charts could be added,  The conclusions can contain more detailed information  about the authors' results

Round 2

Reviewer 1 Report

I would like to thank the authors for having taken into consideration and responded perfectly to practically all the aspects commented on in the extensive review.

There is only one aspect that they claim to have included but which I have not seen reflected in the text and that is point 23 on the Sobel operators. It was only a suggestion, but as they have adjusted it and I have not seen it, I wanted to indicate it again in case it has been forgotten.

On the other hand, there are some minor aspects that I think should be taken into consideration:

I think that the formulas should be revised because in several of them the reference number of the formula appears on the next line instead of being justified on the right.

In equation 1, the spaces between the variable "b" and the "(x,y)" are not necessary, the same happens with the "f" and its corresponding (x,y).

In equation 3 the symbol "*" has been used to indicate the multiplication operation and yet in equations 4, 5 and 6 the symbol "x" has been used for the same operation. I think that it can generate confusion and that the same symbol should always be used and if possible the "*" or even the "·" or simply nothing. Since the symbol "x" can generate doubts.

I think that the whole text should be checked to ensure that there are no double blanks and/or that there are no words together without space between them, as is sometimes the case, for example, in lines: 315, 378, 386, 391, 392, 446, 448, 468, 530, 558, 567 or 588.

In line 457 the units of frequency of the micrometers must go together so that it does not separate in two lines as it happens in the text.

Line 510 should go to the next page.
